# Multimodal C4:
# An Open, Billion-scale Corpus of Images Interleaved with Text

**Wanrong Zhu**♣* **Jack Hessel**♡*
**Anas Awadalla**♠ **Samir Yitzhak Gadre**◇ **Jesse Dodge**♡ **Alex Fang**♠
**Youngjae Yu**† **Ludwig Schmidt**♠♡‡ **William Yang Wang**♣ **Yejin Choi**♠♡
♣University of California, Santa Barbara ♡Allen Institute for Artificial Intelligence
♠Paul G. Allen School of Computer Science, University of Washington
◇Columbia University †Yonsei University ‡LAION
https://github.com/allenai/mmc4

## Abstract

In-context vision and language models like Flamingo [2] support arbitrarily interleaved sequences of images and text as input. This format not only enables few-shot learning via interleaving independent supervised (image, text) examples, but also, more complex prompts involving interaction between images, e.g., "What do image A and image B have in common?" To support this interface, pretraining occurs over web corpora that similarly contain interleaved images+text. To date, however, large-scale data of this form have not been publicly available.

We release Multimodal C4 (mmc4), an augmentation of the popular text-only c4 corpus[2] with images interleaved. We use a linear assignment algorithm to place images into longer bodies of text using CLIP features [24], a process that we show outperforms alternatives. mmc4 spans everyday topics like cooking, travel, technology, etc. A manual inspection of a random sample of documents shows that a vast majority (88%) of images are topically relevant, and that linear assignment frequently selects individual sentences specifically well-aligned with each image (80%). After filtering NSFW images, ads, etc., the resulting mmc4 corpus consists of 101.2M documents with 571M images interleaved in 43B English tokens.

## 1 Introduction

In-context learning [7] enables sequence models to adapt to new tasks without any parameter updates. By interleaving a few supervised examples in a prompt, few-shot learning can be formatted as a next-token prediction task, i.e., $x_1, y_1, x_2, y_2, \ldots, x_n$ is input to predict $\hat{y}_n$. Some image+text models also support in-context learning via interleaving of images/text jointly. Prior experiments [2] suggest that performant multimodal in-context learning is dependent upon pretraining on similarly interleaved sequences of images and text (rather than single image/caption pairs). However, such a large-scale corpus has not been made publicly available.

To address this, we introduce Multimodal C4 (mmc4), a public, billion-scale image-text dataset consisting of interleaved image/text sequences.[3] mmc4 is constructed from public webpages contained in the cleaned English c4 corpus. In addition to standard preprocessing steps like deduplication,

---

*equal contribution; work partly conducted while Wanrong Zhu was an intern at AI2.
[2]https://www.tensorflow.org/datasets/catalog/c4
[3]mmc4's datasheet [15] is available here.

Table 1: Comparison of `mmc4` with other interleaved image/text pretraining corpora. In addition to the full version of the dataset, we also release: 1) fewer-faces subsets, which aim to remove all depicted human faces; and 2) "core" subsets, result from more stringent filtering.

| | # images | # docs | # tokens | Public? |
|---|---|---|---|---|
| M3W (Flamingo) [2] | 185M | 43M | - | ✗ |
| Interleaved training data for CM3 [1] | 25M | 61M | 223B | ✗ |
| Interleaved training data for KOSMOS-1 [17] | ⩽ 355M | 71M | - | ✗ |
| Multimodal C4 (`mmc4`) | 571M | 101.2M | 43B | ✓ |
| Multimodal C4 fewer-faces (`mmc4-ff`) | 375M | 77.7M | 33B | ✓ |
| `mmc4` core (`mmc4-core`) | 29.9M | 7.3M | 2.4B | ✓ |
| `mmc4` core fewer-faces (`mmc4-core-ff`) | 22.4M | 5.5M | 1.8B | ✓ |

NSFW removal, etc., we place images into sequences of sentences by treating each document as an instance of a bipartite linear assignment problem, with images being assigned to sentences (under the constraint that each sentence is assigned at most one image). We show that applying CLIP ViT-L/14 [24] to estimate bipartite weights in a zero-shot fashion results in state-of-the-art performance on intra-document alignment benchmarks, and then apply this process to 100M+ documents to construct `mmc4`. Apart from the full corpus, we have created two additional subsets: `mmc4-ff`, which removes images with detected faces, and `mmc4-core`, a more strictly filtered and downsized version of the corpus, serving as an initial corpus for developers.

We explore `mmc4`, showing that: 1) the text and images in the corpus span expected everyday topics like cooking and travel; 2) filters like NSFW/ad removal work with high accuracy; and 3) the resulting images are relevant to the associated documents, and often, appropriately aligned to the most-relevant individual sentence. We conclude by discussing initial use-cases of `mmc4`, including OpenFlamingo [3],[4] an open source version of Flamingo [2]. Initial ablations show that training on the sequences of `mmc4` enables few-shot, in-context adaptation to image captioning datasets.

## 2 Related Dataset Work

Most million/billion-scale, public multimodal pretraining datasets consist of images paired with their literal descriptions, e.g., LAION-2B [26], CC-12M [8], YFCC100M [32]. However, literal description is only one of many ways images can relate to text on the web [21]. `mmc4` aims to capture a broader range of these relationship types. Some web datasets collect multiple images for one text snippet (e.g., the Google Local Restaurant Reviews Dataset [36] with 4.4M images), or situate images in longer bodies of text (e.g., the Wikipedia-based Image Text Dataset [30] with 11.5M images), but do not directly cover multi-image/multi-sentence interleaving. Table 1 provides summary statistics of other large-scale interleaved pretraining datasets. `mmc4` contains more images than prior non-public datasets. [5] highlight risks associated with web-scale multimodal data.

In addition to the detailed curation steps described in § 3 and the considerations for data release outlined in § 3.1, we are hopeful that the availability of `mmc4` can facilitate a more transparent and critical examination of interleaved corpora compared to previous privately held training sets. Models trained on `mmc4` inherit its risks; we selected the widely-adopted `c4` corpus as a starting point in part because there are existing auditing efforts on the text-only corpus, see § 3 and [23] for more discussion of transparency.

## 3 Data Curation Process

**Initial data collection.** Multimodal C4 is an expansion of the text-only `c4` dataset [25], which was created by taking the April 2019 snapshot from Common Crawl[5] and applying several filters with the intention of retaining high-quality, natural English text. Each document in `c4` consists of the text

---

[4]`https://github.com/mlfoundations/open_flamingo`
[5]`https://commoncrawl.org/`

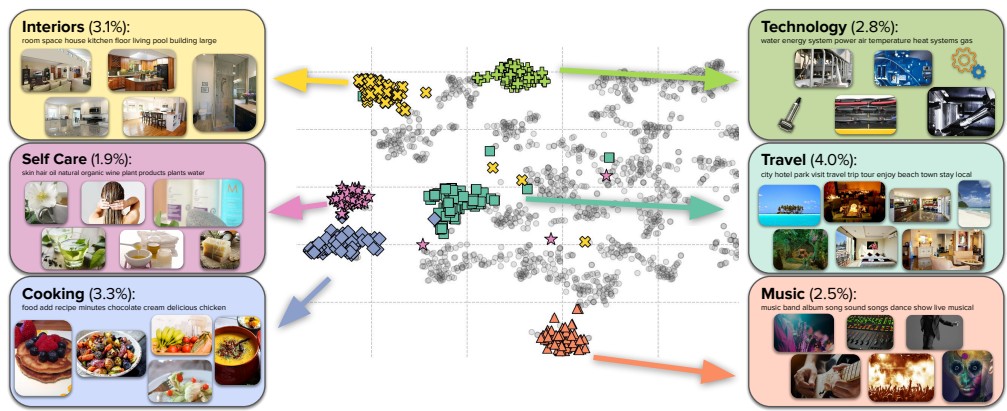

Figure 1: A T-SNE [34] projection of LDA [6] topic clusters from a random sample of 22K documents from `mmc4`; `mmc4` spans a variety of everyday topics, e.g., cooking, technology travel, etc. For 6 selected topics, we also show a sample of most-central images to the topic according to CLIP ViT-L/14 [24].

scraped from one URL. The full `c4` dataset has 365M documents and 156B tokens, covering many domains [12]; it was first used to train T5 [25]. We built the `mmc4` dataset on top of `c4` because: 1) `c4` is a web-scale dataset widely adopted as a pre-training corpus [25, 29, 9, 33, 31]; 2) `c4` is constructed from web pages, which frequently contain multimedia content like images, which makes it a suitable basis for extending to a multimodal sequence version; and 3) `c4-en`,[6] the specific underlying subset from which we construct `mmc4` has already been processed with several data-cleaning steps (including English-language identification by `langdetect`[7] with at least 0.99 confidence; text deduplication removing duplicate three-sentence spans + placeholder text like "lorem ipsum"; and removal of any document containing any word on the "List of Dirty, Naughty, Obscene or Otherwise Bad Words").[8] See [25] for more information about the text-only `c4`. Importantly, by building on the popular text-only `c4`, prior text-only documentation efforts [12] can provide insight about potential biases and risks that could arise when training on our multimodal extension. We use the NLTK [4] sentence tokenizer to chunk each `c4` document into a list of sentences.

**Gathering images.** We first retrieve the original webpages for each document in the `c4-en` dataset from the Common Crawl version `2019-18`, which is the default version for `c4`. Next, we extract the URLs for downloadable images from the raw WAT files. We restrict the image extension to either `png/jpeg/jpg`, and exclude image URLs that contain the following tokens: {`logo`, `button`, `icon`, `plugin`, `widget`}. We attempt to download from these URLs, and resize images to a maximum dimension of 800px. We eliminate any `c4` documents that do not contain valid, downloadable images at the time of collection (mid-to-late 2022). The starting point after this step is 115M documents and 1.37B images.

**De-duplication+small resolution.** We next run duplicate image detection using opennota's `findimagedupes`[9] which uses `phash`[10] to identify visually similar images.[11] We keep only one copy of an image if multiple versions are detected within the same document. We also remove images with more than 10 duplicates in a sample of 60K images. We discard images with a width or height smaller than 150px; this accounts for many small icons, e.g., navigation buttons. We discard images with an aspect ratio of greater than 2 or less than 0.5; this accounts for many banner-like ads. In

---

[6]`https://www.tensorflow.org/datasets/catalog/c4#c4en_default_config`

[7]`https://pypi.org/project/langdetect/`

[8]`https://github.com/LDNOOBW/List-of-Dirty-Naughty-Obscene-and-Otherwise-Bad-Words`

[9]`https://gitlab.com/opennota/findimagedupes`

[10]`http://www.phash.org/`

[11]We use a more aggressive de-duplication threshold of 5 compared to the default library setting of 0; this removes roughly 10M additional images. While some duplicates survive this process, we qualitatively found a threshold of 5 to be an appropriate balance of false positives/negatives.

Table 2: Performance on single document image-text benchmarks from [16] (higher=better in all cases). Applying CLIP ViT-L/14 in a zero-shot fashion [24] produces better within-document alignments compared to prior methods which rely on fine-tuning.

| | MSCOCO | | Story-DII | | Story-SIS | | DII-Stress | | RQA | | DIY | |
|---|---|---|---|---|---|---|---|---|---|---|---|---|
| | AUC | p@1 | AUC | p@1 | AUC | p@1 | AUC | p@1 | AUC | p@1 | AUC | p@1 |
| Random | 49.7 | 5.0 | 49.4 | 19.5 | 50.0 | 19.4 | 50.0 | 2.0 | 49.4 | 17.8 | 49.8 | 6.3 |
| Hessel et al. (2019) [16] | 98.7 | 91.0 | 82.6 | 70.5 | 68.5 | 50.5 | 95.3 | 65.5 | 69.3 | 47.3 | 61.8 | 22.5 |
| Li et al. (2021) [20] | 99.3 | **97.6** | 85.5 | 77.2 | 70.2 | 53.1 | – | – | – | – | – | – |
| CLIP ViT-L/14 (Zero Shot) | **99.4** | 95.7 | **92.8** | **93.9** | **79.1** | **73.3** | **98.7** | **93.0** | **80.7** | **70.7** | **74.0** | **57.6** |

a manual sample of 3.7K images that survive this (and the NSFW) filter, 91 images (2.5%) were identified as ads potentially unrelated to document contents.[12]

**Discarding NSFW images.** We employ strict NSFW image filtering, using DataComp's [14] `dataset2metadata`[13] NSFW binary image classifier. The model is a 4-layer MLP, trained on the NSFW dataset introduced in LAION-2B [26]. This MLP takes as input image features extracted from OpenAI's CLIP ViT-L/14 [24] and achieves 97.4% accuracy on the NSFW test set. We run this classifier on each image and discard cases with a model-predicted NSFW probability over 0.1, which removes approximately 10% of remaining images. Because the data distribution of the classifier and `mmc4` may be slightly different, we also conduct a spot check on images that are marked safe for work. In a manual sample of 3.7K images, we discovered zero NSFW images.

**Aligning images and sentences.** After collecting a set of images for each document, we now describe our intra-document alignment process to interleave the collected images with the sentences. Given that the scope of the images and sentences may be different – the image set is collected from the whole webpage, while the sentence list is subject to preprocessing within the `c4` dataset and thus may not represent the complete content of the webpage – we did not rely on Document Object Model placements in the raw HTML to establish the alignment between images and sentences in each document. Instead, to associate each image with a sentence, we consider each document as an instance of a bipartite assignment problem [19, 16], and use CLIP ViT-L/14 compute pairwise similarities between all sentences/images on a single page. Then, we discard images without at least a 0.15 CLIP cosine similarity to at least one sentence in the document. Finally, we use [18] to compute a bipartite assignment of images to sentences, under the constraint that each sentence can only be assigned a single image.[14] Table 2 shows that this zero-shot application of CLIP ViT-L/14 for within-document matching surpasses prior competitive, fine-tuned methods on image-text alignment benchmarks from [16] (we also distribute the raw intra-document similarity matrices with `mmc4` so alternate assignment methods can be explored). Figure 2 illustrates two example documents with the images interleaved before or after the assigned sentences.

### 3.1 Considerations for data release

`mmc4` contains all images that survive the previously described filters. In addition to the full version of the corpus, we construct two additional types of subsets.

### 3.1.1 Fewer Faces (`mmc4-ff`)

Like the text-only version of `c4`, `mmc4` may contain webpages with personal information that individuals had not explicitly intended to make available for model training. For an initial public release, we make a version of `mmc4` available, `mmc4-ff` (`ff` stands for "fewer faces"); similar to some prior image dataset curation efforts [13, 11], `mmc4-ff` aims to remove images containing detected faces.

---

[12]The delineation between an "irrelevant advertisement" and a "relevant image" is inexact: for example, we discovered images advertising specific, small events, e.g., ones hosted by a fishing club within a city (this type of image was not included in this count). We later assess advertisement-ess in the context of the text of documents, rather than assessing based on the image alone.

[13]https://github.com/mlfoundations/dataset2metadata

[14]For documents with more images than sentences, after assigning an image to each sentence, we assign according to max similarity.

**Example#1**: Interleaving the image *before* each corresponding text

```
[..., "Check out Shane Driscoll's take on sustainable communities and how his photograph fits this year's Green Cities theme.", ..., ,"Man-
made platforms like the one pictured here allow these fish-eating birds of prey to thrive in developed coastal areas.", "A city surrounded by
mountains.", "I took this photo in October on a hike in New Hampshire.", , "It is looking at Mt. Chicora from the middle sister
mountain.", "Getting people out into beautiful places like this is becoming more and more popular, and each time we bring a little piece of nature
back with us that inspires us to make our cities better.", ...]
```

**Example#2**: Interleaving the image *after* each corresponding text

```
["This Walnut and Blue Cheese Stuffed Mushrooms recipe is sponsored by Fisher Nuts.", , "Stuffed mushrooms are an appetizer that always
grabs my attention at a party.", , "If you are a mushroom lover, like me, you probably feel the same.", "The ideas for stuffing mushrooms
are endless, so many combinations to play with, a couple of my personal favorites are these Mediterranean Stuffed Mushrooms and these Spinach and
Toasted Pine Nut Stuffed Mushrooms.", , "Well, you can officially add these Walnut and Blue Cheese Stuffed Mushrooms to my favorites list.",
"The ingredients for the stuffing are simple, which is always best.", ... ]
```

Figure 2: Two example image+text documents from `mmc4`. Following Flamingo [2], during training, images can be interleaved before or after their assigned sentences. More example documents are given in Appendix D.2.

**Removing images with detected faces.**    To detect faces at billion-scale with the intent of removing them from the dataset, we first run RetinaFace[10][15] over a sample of 60K images with the default settings. This detector runs at a high resolution and would be computationally prohibitive to run in full precision for the whole corpus; it produces detailed localization information about the coordinates of each face in each image (which we discard). Using an 80/20 train/test split, we train a cross-validated logistic regression over CLIP ViT-L/14 features to predict whether or not RetinaFace detects a face: this classifier is several orders of magnitude faster compared to RetinaFace. This approximation performs well: we choose a confidence cutoff that achieves 95% recall[16] for the label "RetinaFace detected any face" over the test set while preserving 65% of the original images.

**Manual sample-based face image risk assessment.**    We performed a manual verification of face removal. In a random sample of 912 images that pass all filters including the "no faces" filter, 23 (2.5%) images arguably contain a mostly-un-obscured human face. In most cases (12/23), faces are very low resolution, e.g., a 150x150px image of a crowd of people from a distance, where each face accounts for 3x4 pixels, or are motion shots where the face is blurred. In one case, the face is Marilyn Monroe's as depicted in art on a wall. In 6 cases, there is a plausibly identifiable face depicted: in 2 cases, these are models posing in ads; in 1 case, there is a low resolution image of politicians giving a speech; in 2 cases, the faces are obscured; in 1 case, a passerby was caught in the background of a city photograph and could feasibly be individually identified. Overall: the rate of unobscured, high-resolution, identifiable faces in `mmc4-ff` is low.

### 3.1.2   Core (`mmc4-core`)

Early conversations with some model developers revealed a desire to work with a smaller subset of the corpus as an initial step. We thus additionally release `core` versions of `mmc4` (and `mmc4-ff`), which apply even more stringent filtration criteria. The aim of `core` is to identify a "higher-precision" subset of documents that: 1) have a minimum/maximum number of sentences/images per document; 2) pass an even stricter deduplication step; and 3) have a higher image-text similarity. Hyperparameters[17] are selected heuristically and are balanced to downsize the original corpus by an order of magnitude.

---

[15]As implemented by [27, 28] available from https://github.com/serengil/retinaface.

[16]RetinaFace is not perfectly accurate, so selecting a more aggressive threshold (e.g., 99.99%) would not necessarily result in significantly fewer face-containing images removed.

[17]Min/max number of sentences: 4/40; min/max number of images 2/15; `findimagedupes` applied with a threshold of 10; documents are required to have at least 75% of image assignments have CLIP ViT-L/14 similarity of greater than 25.

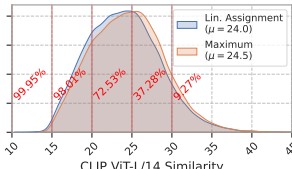 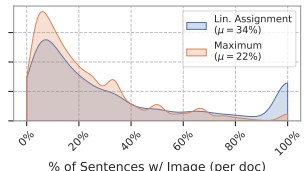 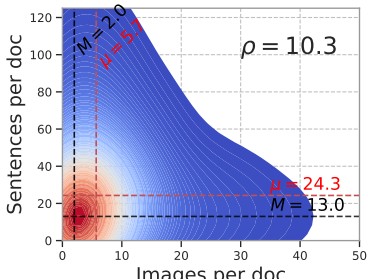

(a) CLIP sim is similar between lin. assignment + max. In red: percent of images remaining at various CLIP thresholds.

(b) Lin. assignment results in a higher percentage of sentences being associated with an image.

Figure 4: Using linear assignment results in comparable image-text similarities to max assignment, but the former spreads images much more evenly, e.g., the per-document mean percent of sentences with an associated image increases from 22% to 34%.

Figure 5: Distribution of images and sentences per document; the median document has 2 images/13 sentences. Documents with more sentences tend to have more images, but the correlation is weak (Spearman $\rho = 10.3$).

## 4 Exploring `mmc4`

**Statistics.** Table 1 gives basic summary statistics of `mmc4` (and fewer-faces/core subsets) compared to some other interleaved image/text corpora. Overall, the full version of `mmc4` is larger than prior non-public datasets across axes like number of images/number of documents. In addition, the various subsets of the corpus offer trade-offs between privacy, image/text similarity thresholds, etc. Figure 5 gives details about the mean/median number of images/sentences in each document (mean/median # sent.=2.0/5.7; # im = 13.0/24.3) based on a random sample of 22K documents.

**Sources of documents & images.** We trace back the top-level domains of documents (webpages) and images to better understand the origins of contents in `mmc4`. Figure 6 presents the top-20 top-level domains that host the highest number of documents and images in `mmc4`. The distribution of document sources in `mmc4` reveals a relatively uniform pattern, with 101.2M documents distributed across 6.0M unique domains. On average, each domain contains approximately 16.9 documents, with a median value of 2.0. The top 10% most frequently appeared domains account for 77% of all documents in `mmc4`. The documents are most commonly hosted on news media outlets (e.g., BBC, NY Times, Daily Express, Daily Mail), academic publication sites (e.g., Springer), online encyclopedias (e.g., Wikipedia), and e-commerce sites (e.g., iTunes, Etsy). Conversely, the sources of images in `mmc4` exhibit a higher level of clustering. The 571.4M images are hosted on 4.9M domains, with each domain having an average of 116.0 images and a median value of 7.0 images. The top 10% most frequent domains are responsible for hosting 89% of all images. Images are most commonly hosted on blogs (e.g., Blogspot, WordPress), shopping sites (e.g., Amazon), cloud storage sites (e.g., AWS S3, Google storage), or general image hosting sites (e.g., Flickr, Imgur). More detailed lists of top document/image domains in `mmc4` and `mmc4-core` can be found in Appendix C.

**Image-text similarity.** Figure 4 provides detail about the linear assignment process compared to a "max" assignment alternative, where each image is simply assigned to its maximally CLIP-similar sentence. The linear assignment process slightly decreases the average CLIP similarity between images/sentences (from 24.5 → 24.0), but significantly more evenly "spreads" images throughout the documents: per-document, the mean percentage of sentences with an associated image rises from 22% → 34%.

**Topic-based assessment.** We ran LDA [6] as implemented by Mallet [22] on a random sample of 22K documents from `mmc4` with $k = 30$ topics. The resulting clusters span a broad set of topics like cooking, communities, travel, music, art, etc. Figure 1 shows some example LDA topic clusters.[19] In

---

[18]Multiple sub-sites may exist within a given domain (e.g., `https://i0.wp.com` and `https://i1.wp.com`). We replace specific patterns such as digits or location acronyms with "*" to cluster these sub-sites together.

[19]A full list of topics and their frequencies according to the model is in Appendix B.

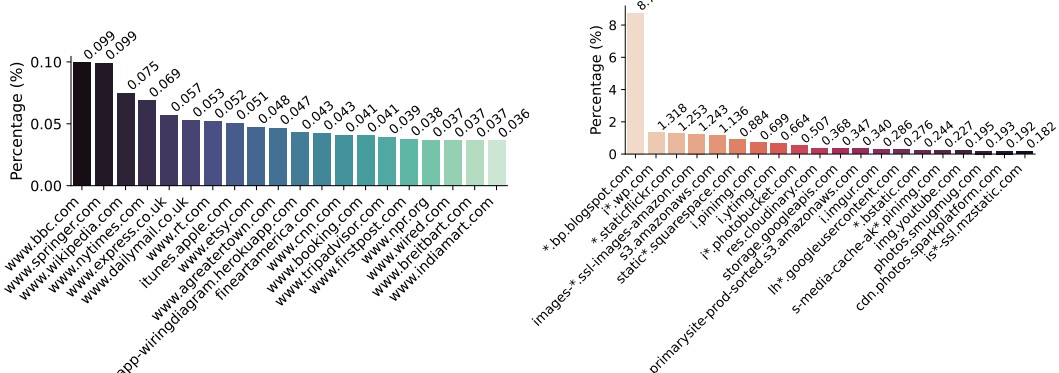

(a) Top-20 most frequent domains for mmc4 documents. (b) Top-20 most frequent domains for images in mmc4.[18]

Figure 6: The top-20 most frequent top-level domains for documents and images in mmc4.

addition, we explore a sample of the images most associated with the corresponding topic,[20] finding that, in general, image topic clusters align with qualitative expectations.

**Manual verification of image relevance+properties.** We randomly sample 200 documents from mmc4 with the goal of assessing how relevant the images contained in the document are to the assigned sentences and to the document as a whole. Table 3 shows the results on the 836 images contained in the 200 documents. 87.7% of all examined images are topically related to the corresponding document, and 80.4% images are well-aligned to the assigned sentences within each document.[21] We also assessed several other factors, finding that: 1) 28.3% contain recognizable human faces; 2) 1.6% contain recognizable watermarks; 3) 3.9% are related to logos;[22] 4) 3.2% are related to advertisements; and 5) 0.7% are duplicated with other images in the same document. Appendix D.1 shows more discussion of images with watermarks, ads/logos, etc.

## 5  OpenFlamingo: An Early Application of mmc4

The first publicly available model to be trained on mmc4 is OpenFlamingo [3]. We run ablations on a small version of OpenFlamingo (3B: backbone = OPT-1.3B [37] language model and CLIP ViT-L/14 [24] vision model) to compare direct training on image captions (LAION-2B [26]) to the interleaved sequences of mmc4-core.[23] To flatten mmc4 documents to training sequences,[24] we: 1) sample a 256 token sub-sequence from each training document; 2) discard images with CLIP image-text similarity less than 20; 3) discard sequences that contain no images after filtering; 4) discard images if there are more than 5 in the resulting sequence.[25] As in [17] we randomly drop sequences with a single image to increase multi-image sequences in the sample.

---

[20]We compute the mean CLIP ViT-L/14 image vector for each topic by associating each image in a document the document's most common topic; then, we compute the mean image vector per topic. Finally, cosine similarity to this mean vector is used to identify the "most topically central" images per-topic.

[21]The alignment between an image and its assigned sentence is a qualitative criterion. We consider an image-sentence pair to be "well-aligned" when the visual elements of the image have a direct and relevant relationship with the text. This can include instances where the image depicts the context or content of the sentence, or where there is a plausible literal overlap between the text and the image, etc.

[22]The logos can be website logos, commercial logos used by businesses or companies to represent their brand or product, or logos for organizations or events. In all cases, the label is assigned if the logo is the primary focus of the image.

[23]These experiments were conducted using a preliminary v1 of the mmc4-core corpus, see this pull request for discussion of small bugfixes in the current v1.1 version.

[24]Future work would be well-suited to investigate the impact of various flattening schemes on downstream performance; the method described here is just one possible method.

[25]Similar to [2], we find that training on a maximum of five image sequences can be sufficient for Open-Flamingo models to generalize to 32 shots during inference.

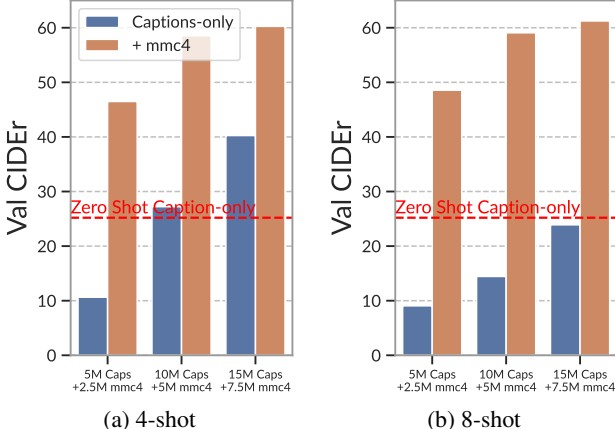

| (a) 4-shot | (b) 8-shot |
|:---:|:---:|

Figure 7: Few shot, in-context MSCOCO captioning performance of OpenFlamingo-3B when training on just captions from LAION-2B vs. mixing in `mmc4-core` sequences. The model trained on `mmc4` sequences is able to generalize to MSCOCO-style captions more effectively vs. the model trained just on LAION-2B image/caption pairs. (Zero shot caption-only=15M caption LAION-2B model)

Table 3: Results of manual verification of 200 randomly sampled documents containing 836 images. A majority of images are topically relevant and well sentence-aligned. The rate of watermarks, ads, duplicates, etc. is low.

|  | % of 836 images |
|---|---|
| Topically-related | 87.7% |
| Sentence-aligned | 80.4% |
| Has face? | 28.3% |
| Has watermark? | 1.6% |
| Logo-related | 3.9% |
| Ads-related | 3.2% |
| Duplicated | 0.7% |

Validation CIDEr [35] results for COCO image captioning are in Figure 7. For 4/8-shot in-context learning settings, the model trained on `mmc4-core` shows 20-30 CIDEr point improvements. The performance of OpenFlamingo-3B trained on just 5M captions/2.5M `mmc4` sequences also exceeds a zero-shot application of OpenFlamingo-3B trained on much more data (15M LAION-2B captions); this provides additional evidence that the interleaving in-context setup enables adaptation to MSCOCO-style captions. The performance of the captions-only OpenFlamingo-3B model degrades from 4-shot to 8-shot learning presumably because these longer sequences are significantly different from the single image/captions it's seen at training time.

## 6    Conclusion

We introduce `mmc4`, a corpus of 100M+ documents with 571M images interleaved in 43B English tokens from the popular `c4` dataset. Initial experimental results show that models trained on image/text sequences from `mmc4` can more effectively perform multimodal in-context learning compared to models trained on single image/captions. We expect interleaving will be important not only for few-shot learning, but also for more diverse multimodal language technologies wherein users may seek to converse with agents with and about visual content in new ways. Future work includes:

1. More precise empirical evaluation of in-context abilities: can models really reason across images/texts in a prompt in flexible ways, or are they limited to interleaved and independent supervised examples?
2. Data scaling: is the performance of in-context vision+language learning bottlenecked by the availability of large-scale interleaved corpora? Or is improved single-modal pretraining sufficient to un-bottleneck multimodal models?
3. Instruction tuning: while interleaving of independent supervised image+text examples enables in-context learning, training an instruction-following multimodal model directly for this case is a promising complementary direction.

## Acknowledgements

We thank the OpenFlamingo team, Sangho Lee, and Jiasen Lu for the helpful discussions, and for being early adopters of `mmc4`. In addition, we thank Jingkang Yang for helpful discussions inspiring `mmc4-core`. We thank Stability AI for the compute for the OpenFlamingo experiments. This work was supported in part by DARPA MCS program through NIWC Pacific (N66001-19-2-4031), the

NSF AI Institute for Foundations of Machine Learning (IFML, CCF-2019844), Open Philanthropy, Google, and the Allen Institute for AI.

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

# A  Dataset Card

Our dataset card is available at `https://github.com/allenai/mmc4/blob/main/DATASET_CARD.md`

# B  Full Set of LDA Topics

Table 4 contains the full set of topics for the $k = 30$ LDA model introduced in § 4.

Table 4: LDA[6] topic modeling outputs (k=30 topics) when trained on a random sample of documents from `mmc4`. Topic frequencies are determined by taking the mean distribution over documents in the corpus. Topic names are generated by GPT-4 conditioned on the top 20 words for each topic, prompted by a request for a short 1-2 word summary.

| Topic name | Rate | Top Words |
| --- | --- | --- |
| E-commerce | 4.61% | products, quality, price, product, online, offer, buy, customers, services, order |
| Healthcare | 2.55% | health, care, body, patients, treatment, medical, pain, cancer, blood, mental |
| Travel | 3.98% | city, hotel, park, visit, travel, trip, tour, enjoy, beach, town |
| Celebrations | 3.94% | fun, wedding, beautiful, christmas, happy, card, birthday, gift, blog, perfect |
| Music | 2.50% | music, band, album, song, sound, songs, dance, show, live, musical |
| Religion | 2.05% | god, church, jesus, lord, faith, man, father, heart, christ, gods |
| Fashion | 4.86% | black, white, size, color, design, wear, style, fabric, cut, fit |
| Nature | 3.05% | water, dog, river, fish, dogs, species, animals, fishing, sea, weather |
| Geography | 3.56% | city, county, state, york, san, north, west, st, john, south |
| Business | 4.15% | management, company, marketing, technology, data, services, team, industry, project, clients |
| Technology | 4.89% | page, app, site, download, website, data, click, google, web, email |
| Education | 2.39% | students, school, learning, skills, children, education, learn, student, training, class |
| Research | 1.43% | data, download, research, analysis, study, al, cells, memory, studies, results |
| Food | 3.31% | food, add, recipe, minutes, chocolate, cream, delicious, chicken, sugar, cheese |
| Law | 2.14% | law, insurance, court, legal, case, state, letter, act, cover, policy |
| Wellness | 1.92% | skin, hair, oil, natural, organic, wine, plant, products, plants, water |
| Self-improvement | 5.27% | change, youre, mind, point, means, fact, thing, ways, question, process |
| Politics | 2.73% | government, president, police, political, war, trump, military, state, party, security |
| Engineering | 2.81% | water, energy, system, power, air, temperature, heat, systems, gas, solar |
| Sports | 3.01% | game, games, team, play, season, players, win, league, player, football |
| Economy | 2.29% | percent, market, million, —, trade, billion, growth, price, company, report |
| Architecture | 3.08% | room, space, house, kitchen, floor, living, pool, building, large, bedroom |
| Automotive | 3.20% | car, vehicle, camera, engine, power, system, model, control, speed, phone |
| Community | 3.91% | community, university, program, research, members, support, development, public, national, group |
| Finance | 1.72% | money, credit, card, real, property, estate, loan, pay, financial, tax |
| International | 2.31% | international, india, countries, china, south, history, united, country, europe, indian |
| Events | 3.93% | 2018, event, pm, 2019, 2017, april, 2016, posted, friday, june |
| Literature | 3.73% | book, story, books, film, series, movie, read, characters, stories, reading |
| Personal | 7.96% | ive, didnt, thing, bit, thought, week, wanted, started, pretty, id |
| Art | 2.70% | art, design, de, images, ikea, image, painting, collection, piano, photo |

# C  Most Frequent Top-Level Domains

Table 5 and Table 6 list the top-50 most frequent top-level domains for documents and images as discussed in § 4. We show domain statistics in both `mmc4` and `mmc4-core`.

Figure 8 shows the top-50 top-level domains of documents in `c4-en` for reference purposes. The domains are sorted by the frequency of occurrence, as the same with results presented in Figure 6. Previous work also discussed the most represented websites in `c4-en` ranked by the total number of tokens [12].

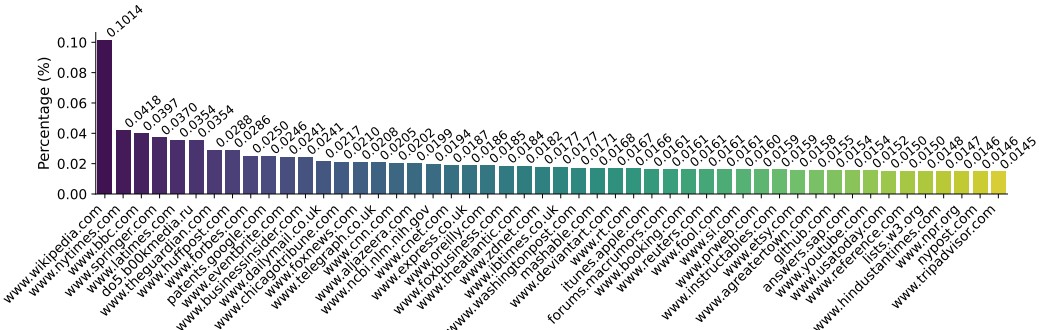

Figure 8: The top-50 most frequent top-level domains for documents in `c4-en`.

# D  Demonstrative Examples

## D.1  Images w/ Watermarks/Ads/Logos

Figure 9a depicts a few sample images containing watermarks in various forms, Figure 9b shows images that are associated with logos, and Figure 9c lists a few sample images related to advertisements. Notice that the dissimilarity between images associated with logos and those pertaining to advertisements is relatively modest. Although images connected to advertisements may occasionally encompass promotional language or persuasive expressions, they may also solely feature logos. Notably, the principal criterion for determining whether an image is ad-related is contingent upon assessing its relevance to the document. If the image is less related to the document, it is more aptly categorized as ad-related. For instance, the interleaved document presented in Table 7 contains two images associated with logos that are intricately linked to the commercial brand being presented within the document. Consequently, these two images are not classified as advertisements.

## D.2  Interleaved Document

Table 7 and Table 8 show two interleaved docs from `mmc4`, displaying the list of sentences and the corresponding assigned images, alongside the CLIP ViT/L-14 image-text similarity score.

Table 5: Top-50 top-level domains for documents in `mmc4` and `mmc4-core`.

| mmc4 documents | | mmc4-core documents | |
|---|---|---|---|
| Domain Name | Percentage | Domain Name | Percentage |
| www.bbc.com | 0.0994% | www.dailymail.co.uk | 0.2352% |
| www.springer.com | 0.0993% | www.alibaba.com | 0.1601% |
| www.wikipedia.com | 0.0750% | www.indiamart.com | 0.1287% |
| www.nytimes.com | 0.0690% | www.teacherspayteachers.com | 0.1116% |
| www.express.co.uk | 0.0573% | www.rt.com | 0.0858% |
| www.dailymail.co.uk | 0.0530% | www.bbc.com | 0.0730% |
| www.rt.com | 0.0519% | www.digit-life.com | 0.0728% |
| itunes.apple.com | 0.0508% | www.cbc.ca | 0.0673% |
| www.etsy.com | 0.0475% | www.stitcher.com | 0.0665% |
| www.agreatertown.com | 0.0468% | local.firestonecompleteautocare.com | 0.0636% |
| app-wiringdiagram.herokuapp.com | 0.0429% | www.monfrague.online | 0.0629% |
| fineartamerica.com | 0.0425% | www.firstpost.com | 0.0555% |
| www.cnn.com | 0.0407% | www.express.co.uk | 0.0552% |
| www.booking.com | 0.0406% | www.androidpolice.com | 0.0535% |
| www.tripadvisor.com | 0.0393% | www.usatoday.com | 0.0528% |
| www.firstpost.com | 0.0377% | www.audible.com | 0.0481% |
| www.npr.org | 0.0368% | itunes.apple.com | 0.0479% |
| www.wired.com | 0.0367% | inhabitat.com | 0.0455% |
| www.breitbart.com | 0.0367% | www.cnn.com | 0.0435% |
| www.indiamart.com | 0.0364% | www.giftacrossindia.com | 0.0433% |
| www.audible.com | 0.0346% | www.houzz.com | 0.0428% |
| medium.com | 0.0342% | appadvice.com | 0.0421% |
| www.dailystar.co.uk | 0.0338% | www.prweb.com | 0.0419% |
| www.weddingwire.com | 0.0336% | www.timeout.com | 0.0414% |
| appadvice.com | 0.0333% | wccftech.com | 0.0412% |
| www.businessinsider.com | 0.0310% | www.ifompt.com | 0.0403% |
| hubpages.com | 0.0303% | phys.org | 0.0383% |
| www.shutterstock.com | 0.0285% | www.abc.net.au | 0.0381% |
| www.alibaba.com | 0.0282% | www.acahome.org | 0.0371% |
| www.techradar.com | 0.0276% | www.npr.org | 0.0368% |
| www.timeout.com | 0.0265% | www.redmondpie.com | 0.0368% |
| economictimes.indiatimes.com | 0.0259% | babyology.com.au | 0.0367% |
| www.prweb.com | 0.0256% | www.etsy.com | 0.0367% |
| www.cbc.ca | 0.0246% | fgontheweb.com | 0.0365% |
| www.houzz.com | 0.0244% | www.pcworld.com | 0.0359% |
| www.ndtv.com | 0.0243% | www.dailystar.co.uk | 0.0350% |
| www.gsmarena.com | 0.0243% | www.realtor.com | 0.0348% |
| gizmodo.com | 0.0243% | www.wikipedia.com | 0.0342% |
| wn.com | 0.0242% | www.advanceduninstaller.com | 0.0342% |
| www.thestar.com | 0.0240% | shopwizion.com | 0.0337% |
| www.deviantart.com | 0.0240% | www.drivermax.com | 0.0337% |
| www.indiebound.org | 0.0238% | www.template.net | 0.0334% |
| www.telegraph.co.uk | 0.0238% | clemsontigers.com | 0.0330% |
| www.teacherspayteachers.com | 0.0236% | www.comparometer.in | 0.0329% |
| www.imdb.com | 0.0234% | maybeloan.com | 0.0320% |
| sg.carousell.com | 0.0233% | medium.com | 0.0320% |
| pixels.com | 0.0228% | shoplionly.com | 0.0320% |
| timesofindia.indiatimes.com | 0.0227% | www.replacement-laptop-battery.com | 0.0314% |
| www.blogtalkradio.com | 0.0227% | www.businessinsider.com.au | 0.0312% |
| www.glamour.com | 0.0223% | www.dummies.com | 0.0312% |

Table 6: Top-50 top-level domains for images in `mmc4` and `mmc4-core`. The symbol "*" is employed to denote specific patterns, such as digits or location acronyms, commonly utilized to differentiate sub-sites within the same domain.

| mmc4 images | | mmc4-core images | |
|---|---|---|---|
| Domain Name | Percentage | Domain Name | Percentage |
| *.bp.blogspot.com | 8.7454% | *.bp.blogspot.com | 6.8934% |
| s3.amazonaws.com | 2.1629% | s3.amazonaws.com | 1.8782% |
| i*.wp.com | 1.3176% | images-*.ssl-images-amazon.com | 1.7976% |
| *.staticflickr.com | 1.2530% | i*.wp.com | 1.4590% |
| images-*.ssl-images-amazon.com | 1.2430% | static*.squarespace.com | 0.9530% |
| static*.squarespace.com | 0.8838% | cdn.atwilltech.com | 0.9009% |
| i.pinimg.com | 0.6992% | *.staticflickr.com | 0.7968% |
| i.ytimg.com | 0.6644% | i.ytimg.com | 0.4446% |
| i*.photobucket.com | 0.5075% | *.imimg.com | 0.4308% |
| res.cloudinary.com | 0.3683% | bt-photos.global.ssl.fastly.net | 0.3827% |
| storage.googleapis.com | 0.3466% | sc*.alicdn.com | 0.3700% |
| i.imgur.com | 0.2858% | i.etsystatic.com | 0.3494% |
| lh*.googleusercontent.com | 0.2762% | i.pinimg.com | 0.3356% |
| *.bstatic.com | 0.2436% | i.dailymail.co.uk | 0.2896% |
| s-media-cache-ak*.pinimg.com | 0.2270% | s-media-cache-ak*.pinimg.com | 0.2705% |
| img.youtube.com | 0.1954% | i.imgur.com | 0.2638% |
| photos.smugmug.com | 0.1934% | i*.photobucket.com | 0.2603% |
| cdn.photos.sparkplatform.com | 0.1915% | lh*.googleusercontent.com | 0.2435% |
| is*-ssl.mzstatic.com | 0.1821% | res.cloudinary.com | 0.2349% |
| i.etsystatic.com | 0.1727% | is*-ssl.mzstatic.com | 0.2142% |
| odis.homeaway.com | 0.1657% | i.bosscdn.com | 0.1989% |
| media-cdn.tripadvisor.com | 0.1605% | assets.eflorist.com | 0.1927% |
| media.karousell.com | 0.1584% | *.yimg.com | 0.1828% |
| www.picclickimg.com | 0.1550% | ecx.images-amazon.com | 0.1356% |
| ae*.alicdn.com | 0.1547% | storage.googleapis.com | 0.1329% |
| m.media-amazon.com | 0.1418% | img.youtube.com | 0.1192% |
| ecx.images-amazon.com | 0.1385% | cdn.shoplightspeed.com | 0.1186% |
| images.furnituredealer.net | 0.1382% | img-aws.ehowcdn.com | 0.1163% |
| image.jimcdn.com | 0.1362% | photos.smugmug.com | 0.1137% |
| bt-photos.global.ssl.fastly.net | 0.1254% | ecdn.teacherspayteachers.com | 0.1047% |
| t.realgeeks.media | 0.1234% | image.jimcdn.com | 0.1035% |
| pbs.twimg.com | 0.1194% | m.media-amazon.com | 0.1006% |
| content.cdntwrk.com | 0.1126% | cdn.webshopapp.com | 0.1000% |
| www.wikihow.com | 0.1106% | i.ebayimg.com | 0.0986% |
| cdn.atwilltech.com | 0.1092% | mediad.publicbroadcasting.net | 0.0915% |
| *.yimg.com | 0.1065% | images.template.net | 0.0906% |
| upload.wikimedia.org | 0.0960% | ae*.alicdn.com | 0.0871% |
| *.media.tumblr.com | 0.0942% | secure.img*-fg.wfcdn.com | 0.0861% |
| f*.bcbits.com | 0.0886% | s*.pcdn.co | 0.0848% |
| f.dvipcdn.com | 0.0848% | st.hzcdn.com | 0.0838% |
| photos*.blogger.com | 0.0833% | assets.simpleviewinc.com | 0.0813% |
| cdn*.weddingwire.com | 0.0822% | fgontheweb.com | 0.0793% |
| static.shareasale.com | 0.0815% | images.navidirect.org | 0.0790% |
| secure.img*-fg.wfcdn.com | 0.0812% | cdni.rt.com | 0.0786% |
| c*.alamy.com | 0.0812% | downloads.intercomcdn.com | 0.0777% |
| usercontent*.hubstatic.com | 0.0810% | gallery.mailchimp.com | 0.0750% |
| sc*.alicdn.com | 0.0803% | slideplayer.com | 0.0690% |
| static.showit.co | 0.0783% | cdn.displays*go.com | 0.0677% |
| i.bosscdn.com | 0.0764% | dta*yqvfnusiq.cloudfront.net | 0.0660% |
| *.imimg.com | 0.0742% | images.clickdealer.co.uk | 0.0644% |

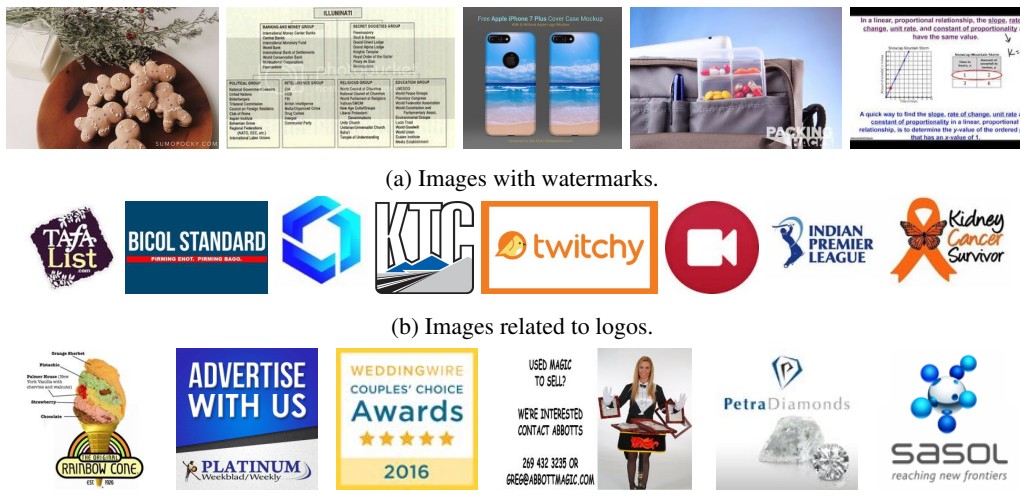

(a) Images with watermarks.

(b) Images related to logos.

(c) Images related to ads.

Figure 9: Manually labeled images with watermarks and images related to logos or ads.

Table 7: An example document from `mmc4` with interleaved sentences and images, together with the CLIP ViT/-14 image-text similarities. This document contains two logo-related images (the 2nd & 3rd images with "NELO") that are relevant to the content of this document, and are therefore excluded from the category of advertisement.

| Sentence | Image | CLIP Similarity |
|---|---|---|
| Our new service for teams to manage their fleets for racing. | | |
| Getting boats has never been this easy. | | |
| Get a step ahead with the planning for your team and get all the boats you need for next season races. | | 23.51 |
| Our new service for teams to manage their fleets for racing. | | 22.40 |
| As easy as adding boats to a list, this service aims to be the simplest way to rent boats, no extra knowledge needed and with full support from our staff. | | |
| Get all the features of a Nelo boat, from having great equipment to our service team for a fraction of the price of a new boat. | | 28.76 |
| All our rental boats for racing are carefully maintained and revised between each race so each boat is as good as new. | | |

Table 8: A document instance retrieved from the mmc4 dataset is presented, consisting of interleaved textual sentences and accompanying images, along with the CLIP ViT/-14 image-text similarity scores.

| Sentence | Image | CLIP Similarity |
|---|---|---|
| Are you thinking about running a retreat for your own group of people? |  | 25.93 |
| We are happy to help you hosting and organizing your own retreat. |  | 19.71 |
| We work with your interest in mind in designing your retreat, and we facilitate the logistics, supporting you all the way for a great experience. | | 21.29 |
| Nestled within powerful and deeply inspiring nature, in the heart of Tuscany, Italy, Podere Di Maggio is a place born of dreams. |  | 22.35 |
| The dream to be close to and learn from nature. |  | 19.37 |
| The dream to create and share beauty. |  | 19.16 |
| The dream to discover and develop the poetry of being and doing. |  | 18.21 |
| We offer an invitation to explore a wide range of life arts:  poetry, dance, music, yoga, meditation, ritual, ceramics, painting, singing, photography, seeing, hearing, touching, feeling, cooking, communicating and collaborating; sharing and daring to discover and unfold yourself. |  | 22.69 |

