## A    Dataset Card

Our dataset card is available at https://github.com/allenai/mmc4/blob/main/DATASET_CARD.md

## B    Full Set of LDA Topics

Table 4 contains the full set of topics for the $k = 30$ LDA model introduced in § 4.

Table 4: LDA[6] topic modeling outputs (k=30 topics) when trained on a random sample of documents from `mmc4`. Topic frequencies are determined by taking the mean distribution over documents in the corpus. Topic names are generated by GPT-4 conditioned on the top 20 words for each topic, prompted by a request for a short 1-2 word summary.

| Topic name | Rate | Top Words |
|---|---|---|
| E-commerce | 4.61% | products, quality, price, product, online, offer, buy, customers, services, order |
| Healthcare | 2.55% | health, care, body, patients, treatment, medical, pain, cancer, blood, mental |
| Travel | 3.98% | city, hotel, park, visit, travel, trip, tour, enjoy, beach, town |
| Celebrations | 3.94% | fun, wedding, beautiful, christmas, happy, card, birthday, gift, blog, perfect |
| Music | 2.50% | music, band, album, song, sound, songs, dance, show, live, musical |
| Religion | 2.05% | god, church, jesus, lord, faith, man, father, heart, christ, gods |
| Fashion | 4.86% | black, white, size, color, design, wear, style, fabric, cut, fit |
| Nature | 3.05% | water, dog, river, fish, dogs, species, animals, fishing, sea, weather |
| Geography | 3.56% | city, county, state, york, san, north, west, st, john, south |
| Business | 4.15% | management, company, marketing, technology, data, services, team, industry, project, clients |
| Technology | 4.89% | page, app, site, download, website, data, click, google, web, email |
| Education | 2.39% | students, school, learning, skills, children, education, learn, student, training, class |
| Research | 1.43% | data, download, research, analysis, study, al, cells, memory, studies, results |
| Food | 3.31% | food, add, recipe, minutes, chocolate, cream, delicious, chicken, sugar, cheese |
| Law | 2.14% | law, insurance, court, legal, case, state, letter, act, cover, policy |
| Wellness | 1.92% | skin, hair, oil, natural, organic, wine, plant, products, plants, water |
| Self-improvement | 5.27% | change, youre, mind, point, means, fact, thing, ways, question, process |
| Politics | 2.73% | government, president, police, political, war, trump, military, state, party, security |
| Engineering | 2.81% | water, energy, system, power, air, temperature, heat, systems, gas, solar |
| Sports | 3.01% | game, games, team, play, season, players, win, league, player, football |
| Economy | 2.29% | percent, market, million, —, trade, billion, growth, price, company, report |
| Architecture | 3.08% | room, space, house, kitchen, floor, living, pool, building, large, bedroom |
| Automotive | 3.20% | car, vehicle, camera, engine, power, system, model, control, speed, phone |
| Community | 3.91% | community, university, program, research, members, support, development, public, national, group |
| Finance | 1.72% | money, credit, card, real, property, estate, loan, pay, financial, tax |
| International | 2.31% | international, india, countries, china, south, history, united, country, europe, indian |
| Events | 3.93% | 2018, event, pm, 2019, 2017, april, 2016, posted, friday, june |
| Literature | 3.73% | book, story, books, film, series, movie, read, characters, stories, reading |
| Personal | 7.96% | ive, didnt, thing, bit, thought, week, wanted, started, pretty, id |
| Art | 2.70% | art, design, de, images, ikea, image, painting, collection, piano, photo |

# C   Most Frequent Top-Level Domains

Table 5 and Table 6 list the top-50 most frequent top-level domains for documents and images as

discussed in § 4. We show domain statistics in both `mmc4` and `mmc4-core`.

Table 5: Top-50 top-level domains for documents in `mmc4` and `mmc4-core`.

| mmc4 documents | | mmc4-core documents | |
|---|---|---|---|
| Domain Name | Percentage | Domain Name | Percentage |
| link.springer.com | 0.0702% | www.dailymail.co.uk | 0.2352% |
| www.nytimes.com | 0.0690% | www.alibaba.com | 0.1601% |
| www.express.co.uk | 0.0573% | dir.indiamart.com | 0.1261% |
| www.dailymail.co.uk | 0.0530% | www.teacherspayteachers.com | 0.1116% |
| www.rt.com | 0.0519% | www.rt.com | 0.0858% |
| itunes.apple.com | 0.0508% | www.digit-life.com | 0.0728% |
| www.etsy.com | 0.0475% | www.cbc.ca | 0.0673% |
| www.agreatertown.com | 0.0468% | www.stitcher.com | 0.0665% |
| app-wiringdiagram.herokuapp.com | 0.0429% | local.firestonecompleteautocare.com | 0.0636% |
| fineartamerica.com | 0.0425% | www.monfrague.online | 0.0629% |
| www.bbc.com | 0.0413% | www.firstpost.com | 0.0555% |
| www.booking.com | 0.0406% | www.express.co.uk | 0.0552% |
| www.tripadvisor.com | 0.0393% | www.androidpolice.com | 0.0535% |
| www.firstpost.com | 0.0377% | traveltips.usatoday.com | 0.0503% |
| www.npr.org | 0.0368% | www.audible.com | 0.0481% |
| www.wired.com | 0.0367% | itunes.apple.com | 0.0479% |
| www.breitbart.com | 0.0367% | inhabitat.com | 0.0455% |
| www.bbc.co.uk | 0.0362% | www.giftacrossindia.com | 0.0433% |
| www.audible.com | 0.0346% | news.bbc.co.uk | 0.0432% |
| medium.com | 0.0342% | www.houzz.com | 0.0428% |
| www.dailystar.co.uk | 0.0338% | appadvice.com | 0.0421% |
| www.weddingwire.com | 0.0336% | www.prweb.com | 0.0419% |
| appadvice.com | 0.0333% | www.timeout.com | 0.0414% |
| www.businessinsider.com | 0.0310% | wccftech.com | 0.0412% |
| hubpages.com | 0.0303% | www.ifompt.com | 0.0403% |
| www.shutterstock.com | 0.0285% | phys.org | 0.0383% |
| www.alibaba.com | 0.0282% | www.abc.net.au | 0.0381% |
| www.techradar.com | 0.0276% | www.acahome.org | 0.0371% |
| rd.springer.com | 0.0266% | www.npr.org | 0.0368% |
| en.wikipedia.org | 0.0266% | www.redmondpie.com | 0.0368% |
| www.timeout.com | 0.0265% | babyology.com.au | 0.0367% |
| economictimes.indiatimes.com | 0.0259% | www.etsy.com | 0.0367% |
| www.prweb.com | 0.0256% | fgontheweb.com | 0.0365% |
| www.cbc.ca | 0.0246% | www.pcworld.com | 0.0359% |
| www.houzz.com | 0.0244% | money.cnn.com | 0.0352% |
| www.ndtv.com | 0.0243% | www.dailystar.co.uk | 0.0350% |
| www.gsmarena.com | 0.0243% | www.realtor.com | 0.0348% |
| gizmodo.com | 0.0243% | www.advanceduninstaller.com | 0.0342% |
| wn.com | 0.0242% | shopwizion.com | 0.0337% |
| www.thestar.com | 0.0240% | www.drivermax.com | 0.0337% |
| www.deviantart.com | 0.0240% | www.template.net | 0.0334% |
| www.indiebound.org | 0.0238% | clemsontigers.com | 0.0330% |
| www.telegraph.co.uk | 0.0238% | www.comparometer.in | 0.0329% |
| www.teacherspayteachers.com | 0.0236% | maybeloan.com | 0.0320% |
| www.imdb.com | 0.0234% | medium.com | 0.0320% |
| sg.carousell.com | 0.0233% | shoplionly.com | 0.0320% |
| pixels.com | 0.0228% | www.replacement-laptop-battery.com | 0.0314% |
| timesofindia.indiatimes.com | 0.0227% | www.businessinsider.com.au | 0.0312% |
| www.blogtalkradio.com | 0.0227% | www.dummies.com | 0.0312% |
| dir.indiamart.com | 0.0226% | abcnews.go.com | 0.0309% |

Table 6: Top-50 top-level domains for images in `mmc4` and `mmc4-core`. The symbol "`*`" is employed to denote specific patterns, such as digits or location acronyms, commonly utilized to differentiate sub-sites within the same domain.

| mmc4 images | | mmc4-core images | |
|---|---|---|---|
| Domain Name | Percentage | Domain Name | Percentage |
| *.bp.blogspot.com | 8.7454% | *.bp.blogspot.com | 6.8934% |
| i*.wp.com | 1.3176% | images-*.ssl-images-amazon.com | 1.7976% |
| *.staticflickr.com | 1.2530% | i*.wp.com | 1.4590% |
| images-*.ssl-images-amazon.com | 1.2430% | s3.amazonaws.com | 1.1683% |
| s3.amazonaws.com | 1.1356% | static*.squarespace.com | 0.9530% |
| static*.squarespace.com | 0.8838% | cdn.atwilltech.com | 0.9009% |
| i.pinimg.com | 0.6992% | *.staticflickr.com | 0.7968% |
| i.ytimg.com | 0.6644% | i.ytimg.com | 0.4446% |
| i*.photobucket.com | 0.5075% | *.imimg.com | 0.4308% |
| res.cloudinary.com | 0.3683% | bt-photos.global.ssl.fastly.net | 0.3827% |
| storage.googleapis.com | 0.3466% | sc*.alicdn.com | 0.3700% |
| primarysite-prod-sorted.s3.amazonaws.com | 0.3402% | i.etsystatic.com | 0.3494% |
| i.imgur.com | 0.2858% | i.pinimg.com | 0.3356% |
| lh*.googleusercontent.com | 0.2762% | i.dailymail.co.uk | 0.2896% |
| *.bstatic.com | 0.2436% | s-media-cache-ak*.pinimg.com | 0.2705% |
| s-media-cache-ak*.pinimg.com | 0.2270% | i.imgur.com | 0.2638% |
| img.youtube.com | 0.1954% | i*.photobucket.com | 0.2603% |
| photos.smugmug.com | 0.1934% | lh*.googleusercontent.com | 0.2435% |
| cdn.photos.sparkplatform.com | 0.1915% | res.cloudinary.com | 0.2349% |
| is*-ssl.mzstatic.com | 0.1821% | is*-ssl.mzstatic.com | 0.2142% |
| i.etsystatic.com | 0.1727% | i.bosscdn.com | 0.1989% |
| odis.homeaway.com | 0.1657% | assets.eflorist.com | 0.1927% |
| media-cdn.tripadvisor.com | 0.1605% | *.yimg.com | 0.1828% |
| media.karousell.com | 0.1584% | ecx.images-amazon.com | 0.1356% |
| www.picclickimg.com | 0.1550% | storage.googleapis.com | 0.1329% |
| ae*.alicdn.com | 0.1547% | img.youtube.com | 0.1192% |
| m.media-amazon.com | 0.1418% | cdn.shoplightspeed.com | 0.1186% |
| ecx.images-amazon.com | 0.1385% | img-aws.ehowcdn.com | 0.1163% |
| images.furnituredealer.net | 0.1382% | photos.smugmug.com | 0.1137% |
| image.jimcdn.com | 0.1362% | ecdn.teacherspayteachers.com | 0.1047% |
| bt-photos.global.ssl.fastly.net | 0.1254% | image.jimcdn.com | 0.1035% |
| t.realgeeks.media | 0.1234% | m.media-amazon.com | 0.1006% |
| pbs.twimg.com | 0.1194% | cdn.webshopapp.com | 0.1000% |
| content.cdntwrk.com | 0.1126% | i.ebayimg.com | 0.0986% |
| www.wikihow.com | 0.1106% | mediad.publicbroadcasting.net | 0.0915% |
| cdn.atwilltech.com | 0.1092% | images.template.net | 0.0906% |
| *.yimg.com | 0.1065% | ae*.alicdn.com | 0.0871% |
| upload.wikimedia.org | 0.0960% | secure.img*-fg.wfcdn.com | 0.0861% |
| *.media.tumblr.com | 0.0942% | s*.pcdn.co | 0.0848% |
| f*.bcbits.com | 0.0886% | st.hzcdn.com | 0.0838% |
| f.dvipcdn.com | 0.0848% | assets.simpleviewinc.com | 0.0813% |
| photos*.blogger.com | 0.0833% | fgontheweb.com | 0.0793% |
| cdn*.weddingwire.com | 0.0822% | images.navidirect.org | 0.0790% |
| static.shareasale.com | 0.0815% | cdni.rt.com | 0.0786% |
| secure.img*-fg.wfcdn.com | 0.0812% | downloads.intercomcdn.com | 0.0777% |
| c*.alamy.com | 0.0812% | gallery.mailchimp.com | 0.0750% |
| usercontent*.hubstatic.com | 0.0810% | slideplayer.com | 0.0690% |
| sc*.alicdn.com | 0.0803% | cdn.displays*go.com | 0.0677% |
| static.showit.co | 0.0783% | dta*yqvfnusiq.cloudfront.net | 0.0660% |
| i.bosscdn.com | 0.0764% | images.clickdealer.co.uk | 0.0644% |

# D Demonstrative Examples

## D.1 Images w/ Watermarks/Ads/Logos

Figure 8a depicts a few sample images containing watermarks in various forms, Figure 8b shows images that are associated with logos, and Figure 8c lists a few sample images related to advertise-ments. Notice that the dissimilarity between images associated with logos and those pertaining to

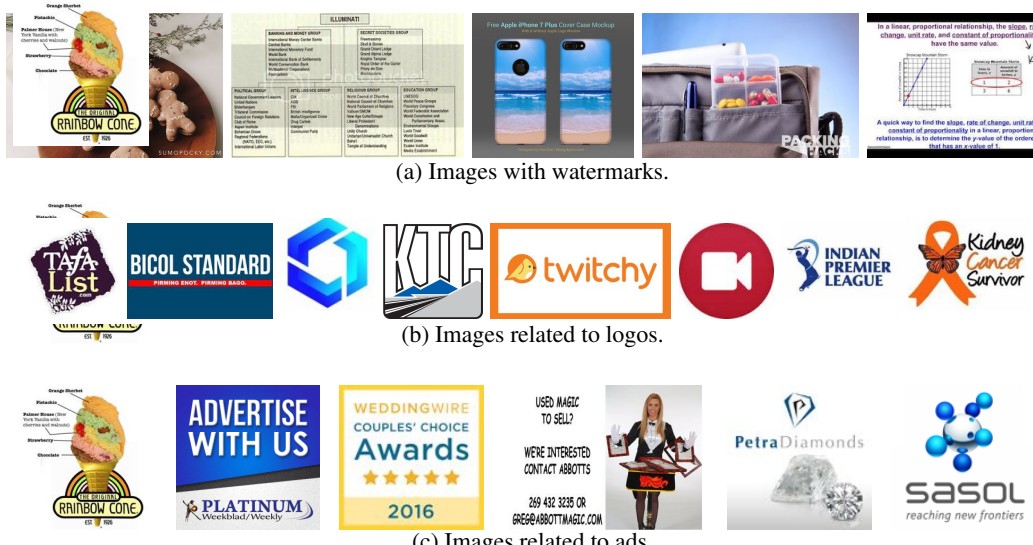

(a) Images with watermarks.

(b) Images related to logos.

(c) Images related to ads.

Figure 8: Manually labeled images with watermarks and images related to logos or ads.

Table 7: An example document from `mmc4` with interleaved sentences and images, together with the CLIP ViT/-14 image-text similarities. This document contains two logo-related images (the 2nd & 3rd images with "NELO") that are relevant to the content of this document, and are therefore excluded from the category of advertisement.

| Sentence | Image | CLIP Similarity |
| --- | --- | --- |
| Our new service for teams to manage their fleets for racing. | | |
| Getting boats has never been this easy. | | |
| Get a step ahead with the planning for your team and get all the boats you need for next season races. | | 23.51 |
| Our new service for teams to manage their fleets for racing. | | 22.40 |
| As easy as adding boats to a list, this service aims to be the simplest way to rent boats, no extra knowledge needed and with full support from our staff. | | |
| Get all the features of a Nelo boat, from having great equipment to our service team for a fraction of the price of a new boat. | | 28.76 |
| All our rental boats for racing are carefully maintained and revised between each race so each boat is as good as new. | | |

advertisements is relatively modest. Although images connected to advertisements may occasionally encompass promotional language or persuasive expressions, they may also solely feature logos. Notably, the principal criterion for determining whether an image is ad-related is contingent upon assessing its relevance to the document. If the image is less related to the document, it is more aptly categorized as ad-related. For instance, the interleaved document presented in Table 7 contains two images associated with logos that are intricately linked to the commercial brand being presented within the document. Consequently, these two images are not classified as advertisements.

## D.2 Interleaved Document

Table 7 and Table 8 show two interleaved docs from `mmc4`, displaying the list of sentences and the corresponding assigned images, alongside the CLIP ViT/L-14 image-text similarity score.

Table 8: A document instance retrieved from the `mmc4` dataset is presented, consisting of interleaved textual sentences and accompanying images, along with the CLIP ViT/-14 image-text similarity scores.

| Sentence | Image | CLIP Similarity |
|---|---|---|
| Are you thinking about running a retreat for your own group of people? |  | 25.93 |
| We are happy to help you hosting and organizing your own retreat. |  | 19.71 |
| We work with your interest in mind in designing your retreat, and we facilitate the logistics, supporting you all the way for a great experience. |  | 21.29 |
| Nestled within powerful and deeply inspiring nature, in the heart of Tuscany, Italy, Podere Di Maggio is a place born of dreams. |  | 22.35 |
| The dream to be close to and learn from nature. |  | 19.37 |
| The dream to create and share beauty. |  | 19.16 |
| The dream to discover and develop the poetry of being and doing. |  | 18.21 |
| We offer an invitation to explore a wide range of life arts: poetry, dance, music, yoga, meditation, ritual, ceramics, painting, singing, photography, seeing, hearing, touching, feeling, cooking, communicating and collaborating; sharing and daring to discover and unfold yourself. |  | 22.69 |