# OpenReview forum: "Multimodal C4: An Open, Billion-scale Corpus of Images Interleaved with Text"
_NeurIPS.cc/2023/Track/Datasets_and_Benchmarks — NeurIPS 2023 Datasets and Benchmarks Poster_

### Official Review · Reviewer_eWRn · 2023-07-20
**A valuable dataset**

**Rating:** 8
**Confidence:** 4
**Clarity:** Yes. The paper is easy to follow.

**Strengths:**

1. The scale of the constructed dataset surpasses that of previous datasets, making it a valuable resource for future multimodality model research and development.
2. The paper provides comprehensive insights into data collection, filtering, and alignment strategies, offering guidance for constructing similar datasets in future studies.
3. The authors have created different subsets of the dataset and established a baseline method, adding versatility to its potential applications.

**Additional Feedback:**

See the weaknesses and limitations.

**Correctness:**

Yes. There is a wealth of information available on data collection, filtering, and alignment strategies.

**Documentation:**

Yes. The supplementary material provides the dataset document.

**Ethics:**

See the comments on the "Limitations" part.

**Limitations:**

Authors should address the social impact of the collected dataset and its limitations. For instance, they should consider whether the model trained on the dataset may potentially leak private information or exhibit biases towards specific groups of people or values.

**Opportunities For Improvement:**

1. Figure 7 demonstrates that utilizing more samples from the mmc4-core dataset leads to continuous performance improvement. However, the paper lacks a discussion on the use of the full dataset for training. It is crucial to investigate the impact of incorporating relatively large-scale but potentially lower quality data on model training.
2. The paper only presents caption results, whereas it would be beneficial to include additional results showcasing in-context learning examples. This would further emphasize the importance of constructing interleaving multimodality datasets.


**Relation To Prior Work:**

Yes. Sec. 2 has presented the related works.

**Summary And Contributions:**

This paper introduces the MultiModal C4 (mmc4) dataset, which consists of interleaved text and vision data. The dataset covers diverse topics and contains images that have a high likelihood of being relevant to the text descriptions. The mmc4 dataset is significantly larger than previous multimodal datasets and holds great potential for advancing the development of multimodality models.

---

> ### Author Response · Authors · 2023-08-17
> **Rebuttal Response to Reviewer eWRn**
>
> Thank you for your detailed reading and valuable insights concerning the mmc4 dataset!  Here are our responses to your questions:
>
> > **Social impact of the collected dataset and its limitations**
>
> As with any dataset that contains data scraped from the web, mmc4 contains information posted on public websites by creators on the Internet. Even though we choose to build mmc4 on top of the widely-used C4 dataset, which has undergone rounds of screening and cleaning, we have to note it here that, at this scale, there is likely factually incorrect information, toxic language, and personal information in the dataset that users of the Internet chose to upload publicly. In our data cleaning process, we have applied image filters for not-safe-for-work contents and human faces. However, it is currently infeasible to detect and remove all instances of problematic data for a dataset at web-scale. We can further acknowledge this in the paper.
>
> We would also like to note that mmc4 is intended as a research artifact. By beginning to train and evaluate models on mmc4, we believe researchers will be able to pose many important questions surrounding bias and fairness in the context of training with interleaved image-text data.
>
> > **Discussion on using full dataset for training**
>
> We trained for as many iterations as we were able --- a full pretraining over 500M+ was prohibitive. We're hopeful that our scaling results at 3 dataset scales provide enough information to suggest that training on the whole corpus is a helpful step.
>
> > **Additional results showcasing in-context learning examples**
>
> We will add a forward citation to works that have used mmc4 as a pretraining corpus and are able to conduct in-context learning successfully, including but not limited to OpenFlamingo v2 [1], Otter [2], and Emu [3].
>
> Our experiments corroborate the claim in the original DeepMind Flamingo paper [4]: *“removing the interleaved image-text dataset results in important decreases in final scores on all tasks from the validation subsets multi-benchmark”*. Such claim is further supported by the authors of Otter in their recent experiments [5]. They found that pretraining their MPT-based model with MMC4+Laion2B outperformed a newer version of Llama-v2-based model being adapter-tuned with CC3M. They noted that pretraining with the interleaved data in mmc4 is crucial to model performance: *“...comparing it with our previous OTTER-MPT7B, we found that adding a more powerful LLM (Llama2) didn't improve performance. This could be due to the lack of interleaved data and visual language alignment pre-training in the Llama2 version.”*
>
> We would like to note that curating and documenting this dataset has been a substantial endeavor. Our primary contribution lies in the creation of the mmc4 dataset itself, rather than in presenting exhaustive experimental results for various models trained on it. We defer more comprehensive experimentation to future research efforts.
>
> [1] https://arxiv.org/abs/2308.01390
>
> [2] https://arxiv.org/abs/2305.03726
>
> [3] https://arxiv.org/abs/2307.05222
>
> [4] https://arxiv.org/pdf/2204.14198v1.pdf (Section 4.4.1)
>
> [5] https://twitter.com/BoLi68567011/status/1684028048998559747

---

### Official Review · Reviewer_y4Co · 2023-07-20
**Review on Multimodal C4: An Open, Billion-scale Corpus of Images Interleaved with Text**

**Rating:** 7
**Confidence:** 4
**Correctness:** It is sound.
**Clarity:** The paper is well written and easy to…

**Strengths:**

1. It provides a pretty large-scale image-text interleaved dataset, which fills the blank of this kind of dataset for multi-modal in-context learning.
2. It provides sufficient data cleaning, simple and effective image-text alignment for improving the quality of the dataset, and provide different subsets of the dataset with different sizes.
3. An initial experiment of OpenFlamingo on MSCOCO image captioning validates the effectiveness of the proposed mmc4 dataset for few-shot learning.

**Additional Feedback:**

I think the dataset is valuable, and It is better if more experiment results on different VLP tasks, different subsets, and zero-shot performance can all be provided, except for only the MSCOCO image captioning task. How the model trained on this dataset performs on video-related tasks which may also have image-text interleaved scenario?

**Documentation:**

Yes

**Opportunities For Improvement:**

1. More thorough experiment on the effectiveness of the dataset on different VLP tasks, different subsets, and also zero-shot performance. It would be better to provide the performance of data scaling in different subsets, and performance on different VLP tasks. And how is the performance for zero-shot performance compared with dataset of image-text pairs?
2. Provide more up-to-date subset of the dataset.


**Relation To Prior Work:**

Yes

**Summary And Contributions:**

This paper releases a first large-scale image-text interleaved dataset for promoting the development of multi-modal in-context learning and few-shot learning. It is built on text-only c4 corpus, and a linear assignment algorithm is used to place images into longer bodies of text using CLIP features. An initial experiment of OpenFlamingo on MSCOCO image captioning validates the effectiveness of the proposed mmc4 dataset for few-shot learning.

---

> ### Author Response · Authors · 2023-08-17
> **Rebuttal Response to Reviewer y4Co**
>
> Thank you very much for your thoughtful comments and suggestions regarding our work! Here are our responses to your questions:
>
> > **More thorough experimentation**
>
> We will add a forward citation to works that have used mmc4 as a pretraining corpus successfully and have reported experimental results on various VLP tasks, including but not limited to OpenFlamingo v2 [1], Otter [2], and Emu [3].
>
> Our experiments corroborate the claim in the original DeepMind Flamingo paper [4]: *“removing the interleaved image-text dataset results in important decreases in final scores on all tasks from the validation subsets multi-benchmark”*. Such claim is further supported by the authors of Otter in their recent experiments [5]. They found that pretraining their MPT-based model with MMC4+Laion2B outperformed a newer version of Llama-v2-based model being adapter-tuned with CC3M. They noted that pretraining with the interleaved data in mmc4 is crucial to model performance: *“...comparing it with our previous OTTER-MPT7B, we found that adding a more powerful LLM (Llama2) didn't improve performance. This could be due to the lack of interleaved data and visual language alignment pre-training in the Llama2 version.”*
>
> We would like to note that curating and documenting this dataset has been a substantial endeavor. Our primary contribution lies in the creation of the mmc4 dataset itself, rather than in presenting exhaustive experimental results for various models trained on it. We defer more comprehensive experimentation to future research efforts.
>
> [1] https://arxiv.org/abs/2308.01390
>
> [2] https://arxiv.org/abs/2305.03726
>
> [3] https://arxiv.org/abs/2307.05222
>
> [4] https://arxiv.org/pdf/2204.14198v1.pdf (Section 4.4.1)
>
> [5] https://twitter.com/BoLi68567011/status/1684028048998559747
>
> > **More up-to-date text vs. C4**
>
> The C4 dataset was created from the April 2019 CommonCrawl snapshot, so it includes text up until 2019. There isn't more up-to-date text in the C4 dataset for us to subset, and C4 is still one of the most common datasets used for language modeling, so we believe it is still relevant for our work. We leave it up to future work to start with a different text corpus that includes more recent data.

---

### Official Review · Reviewer_5HBh · 2023-07-21
**Highly valuable and useful for multimodal learning.**

**Rating:** 10
**Confidence:** 4
**Correctness:** Yes.
**Clarity:** Yes.

**Strengths:**

Scale of the dataset
Approach used for collecting and assigning

**Additional Feedback:**

No additional feedbacks.

**Documentation:**

Yes.

**Ethics:**

No.

**Limitations:**

One of the limitation is about validity of the dataset. It highly depends on the accuracy and validity of the CLIP model.

**Opportunities For Improvement:**

It will be very useful if they can also release the respective scores of the CLIP for each pair. On the other hand, also will be useful if they can add relative information about positional relation of the images and text in the HTML DOM as well.

**Relation To Prior Work:**

Yes.

**Summary And Contributions:**

Authors provided a huge set of image-text dataset that is interleaved. They used CLIP features for assignment. Manual inspection also shows validity of the results as well. Threshold for CLIP (0.15) is a bit questionable, because it might yield into very dissimilar images being assigned. Also single sentence assignment is another questionable part, because it can be assigned for multiple sentences from different parts or even paragraphs, but they decided to assign to single sentences.
In overall, their scheme is very useful along with the dataset for a wide variety of the tasks.

---

> ### Author Response · Authors · 2023-08-17
> **Rebuttal Response to 5HBh**
>
> We are deeply grateful for your comprehensive review and favorable assessment of our paper!
>
> > **Can you release CLIP scores of each image/text pair?**
>
> We did! They are available at the linked repo: https://github.com/allenai/mmc4/
>
> > **What about HTML DOM?**
>
> We did not rely on the HTML DOM for our cross-modality alignment process due to the nature of our mmc4 data collection. The text within mmc4 originates from C4 and has undergone preprocessing. As a result, the text covers only a fraction of the original content present in each raw HTML page. Meanwhile, we crawled all downloadable images for each HTML page. Relying solely on DOM information to identify the nearest text for each image might lead to significant mismatches, as the associated paragraphs might not be encompassed within the C4 text.
>
> Moreover, we try to associate at most one image with each sentence in mmc4. This objective cannot be met through DOM usage, as images can be clustered within albums, potentially leading to multiple images being linked to the same nearest sentence.

---

### Official Review · Reviewer_VvxB · 2023-07-24
**Review of multimodal C4: an open, billion-scale corpus of images interleaved with text**

**Rating:** 7
**Confidence:** 4

**Strengths:**

Compared with other existing interleaved image/text pretraining corpora, mmc4 has a larger scale of images and documents. More importantly, mmc4 is one of the public, large-scale, and cleaned datasets, consisting of interleaved image/text sequences. Moreover, it has been used to train multimodal model OpenFlamingo, and has preliminarily verified its effectiveness.

**Additional Feedback:**

See above.

**Clarity:**

The paper is well-written overall, but there are still some issues that need to be addressed:
(1) Acronym usage: The paper should use the full name when using an acronym for the first time, so that the readers can understand it. For example, NSFW (Not Safe For Work) in the abstract and LDA (Latent Dirichlet Allocation) in Sec.4.
(2) Paper structure: The Sec.3 of the paper only contains Sec.3.1, which does not conform to the conventional paper structure.
(3) Key information missing: Some key information is not given in detail in the paper, making it difficult for the readers to follow the paper’s logic. For example, in Sec.3.1.2, what is “stricter deduplication step”? What is the y-axis of Fig.4? These information should be clearly given in the manuscript.
(4) grammar mistakes: There are some grammatical errors exist in the manuscript. singular/plural misuse, Article or Preposition omission (e.g., “example” should use the plural in line 178, “the” is missing in line 39, “to” is missing in line 108)

**Correctness:**

The construction of the dataset is reasonable and considerate. The author tries to extend the variety of the dataset.

**Documentation:**

The authors provide detailed details on data collection and processing, as well as the URL for obtaining the dataset

**Ethics:**

I think that this manuscript does not have any ethical issues, because the mmc4 method proposed in this manuscript uses a public dataset C4 and excludes the images with faces in the released version, does not involve any unethical or illegal application scenarios, and does not violate any copyright and intellectual property laws. Therefore, I think that this manuscript is qualified in terms of ethics, and does not need further discussion or review.

**Limitations:**

The authors have addressed the privacy concerns in the data collection and construction process, and have made available mmc4-ff and mmc4-core-ff, which filter out images with faces. However, the reliability of using RetinaFace to detect and remove face images is questionable, as it may fail to delete some images with faces. Therefore, it would be advisable to perform manual or additional screening.

**Opportunities For Improvement:**

1. The role of the dataset has not been fully validated. This manuscript only verifies the image caption performance of OpenFlamingo, which is trained on mmc4. Moreover, the effectiveness of this work needs to be validated on other models besides OpenFlamingo.
2. The manuscript lacks some necessary experimental results that need to be presented. (e.g., ”Ablations on a small version of OpenFlamingo” in Sec.5.)
3. The results of the manual verification mentioned in Sec.4 have not been fully validated. This manuscript randomly sampled 200 documents containing 836 images to verify the relevance between images, sentences, and documents. However, the mmc4 dataset proposed in this manuscript consists of 101.2M documents, and the sampled number of documents only accounts for 0.0002% of the total.

**Relation To Prior Work:**

Yes.

**Summary And Contributions:**

To address the current lack of a publicly large-scale corpus that contains similarly interleaved sequences of images and text, this manuscript extends the text-only C4 dataset by applying Gathering, Filter, and Aligning processes to construct a multimodal dataset mmc4，which consists of 101.2M documents with 571M images interleaved in 43B English tokens. What’s more, a publicly available model OpenFlamingo is trained on mmc4 and achieve promising results. This manuscript presents the details of dataset construction and discusses the potential applications and challenges of multimodal research.

---

> ### Author Response · Authors · 2023-08-17
> **Rebuttal Response to Reviewer VvxB**
>
> We appreciate your insightful comments and questions! Your feedback helps us to address important aspects of our work that deserve more detailed consideration. Here's our response to your observations.
>
> > **Only 836 images were examined, how can you be sure of the quality?**
>
> Because the documents were sampled uniformly at random, we can compute 95% confidence intervals for the whole corpus, even though we only examine a small portion of it. Specifically, here are the 95% confidence intervals for the statistics in Table 3, which we can add to the main paper.
> |                   | Low   | High  |
> |-------------------|-------|-------|
> | Topically-related | 85.5% | 89.9% |
> | Sentence-aligned  | 77.7% | 83.1% |
> | Has face?         | 25.3% | 31.4% |
> | Has watermark?    | 0.7%  | 2.4%  |
> | Logo-related      | 2.6%  | 5.3%  |
> | Ads-related       | 2.0%  | 4.4%  |
> | Duplicated        | 0.1%  | 1.3%  |
>
>
> > **Manual verification of RetinaFace accuracy**
>
> In L131, we describe a manual verification of our face discarding filter, including detailing error cases like photos of famous people, or faces that are in the background and account for only ~12 pixels. A manual examination of a random sample of N=912 images reveals 6 images with unobscured faces, which yields a 95% confidence interval of (0.1%, 1.2%) for the whole corpus.
>
> We agree that RetinaFace is not a perfect filter. However, it is currently infeasible to detect and remove all instances of problematic data for a dataset at web-scale with pure human effort. Unfortunately, this is a shared challenge and shortcoming of almost all web-scale datasets. Please let us know if you have any suggestions on the choice of automatic filtering toolkit, and we would be happy to follow your advice and conduct stricter human face filtering if possible.
>
>
> > **Additional experimental results besides OpenFlamingo**
>
> We will add a forward citation to works that have used mmc4 as a pretraining corpus successfully, including but not limited to OpenFlamingo v2 [1], Otter [2], and Emu [3].
>
> Our experiments corroborate the claim in the original DeepMind Flamingo paper [4]: *“removing the interleaved image-text dataset results in important decreases in final scores on all tasks from the validation subsets multi-benchmark”*. Such claim is further supported by the authors of Otter in their recent experiments [5]. They found that pretraining their MPT-based model with MMC4+Laion2B outperformed a newer version of Llama-v2-based model being adapter-tuned with CC3M. They noted that pretraining with the interleaved data in mmc4 is crucial to model performance: *“...comparing it with our previous OTTER-MPT7B, we found that adding a more powerful LLM (Llama2) didn't improve performance. This could be due to the lack of interleaved data and visual language alignment pre-training in the Llama2 version.”*
>
>
> We would like to note that curating and documenting this dataset has been a substantial endeavor. Our primary contribution lies in the creation of the mmc4 dataset itself, rather than in presenting exhaustive experimental results for various models trained on it. We defer more comprehensive experimentation to future research efforts.
>
> [1] https://arxiv.org/abs/2308.01390
>
> [2] https://arxiv.org/abs/2305.03726
>
> [3] https://arxiv.org/abs/2307.05222
>
> [4] https://arxiv.org/pdf/2204.14198v1.pdf (Section 4.4.1)
>
> [5] https://twitter.com/BoLi68567011/status/1684028048998559747
>
>
> > **Improving manuscript clarity**
>
> Thanks for pointing out the issues in acronym usage, section structure and grammar! We apologize for the oversight and will fix the writing issues in the revision. Regarding your questions:
>
> * In Sec.3.1.2, the “stricter deduplication step” refers to raising the de-duplication threshold of *findimagedupes* from 5 (for mmc4) to 10 (for mmc4-core). The hyperparameter setting is mentioned in footnote 11 and footnote 17 in the submission, and we will further clarify this in the revision.
>
> * The two subfigures in Figure 4 are kernel density estimate plots that visualize the data distribution. Here, the y-axis refers to the “density”.
>
> We will follow your suggestion and add further clarification in the revision.

---

### Official Review · Reviewer_GHE4 · 2023-07-24

**Rating:** 7
**Confidence:** 3
**Correctness:** Cirrect
**Clarity:** Clear

**Strengths:**

Using the bipartite mapping problem to map a set of images to a set of text makes sense. I like it. The approach is also scalable because of its simplicity.

The amount of data in the proposed dataset is huge. Also, the above mapping allows for a relatively accurate correspondence between images and text.

The description of the construction and cleaning of the dataset is detailed. Such a description is helpful for researchers who will later try to build such a dataset.


**Additional Feedback:**

None

**Documentation:**

Enough

**Ethics:**

No problem

**Limitations:**

This manuscript is significantly over the page limit. The excess is in the supplementary file itself. In other words, the authors mistakenly attached the supplemental file to the main body of the manuscript.

According to the review guidelines, a few extra lines should not be a problem [1]. Also, the excesses in this manuscript can be easily separated by camera-ready. But in any case, I don't think that a paper should be given a higher score if it doesn't follow the basic rule of page excesses. Therefore, I give this paper a score of 7 instead of 8-10.

[1] "In general, we were lenient with minor formatting violations (e.g., a spillover to page 10), as long as these violations can be easily rectified in the final version." from https://nips.cc/Conferences/2023/DatasetsAndBenchmarks/ReviewGuidelines

**Opportunities For Improvement:**

The data curation process proposed in this paper is very detailed and thoughtful. It would be a further contribution to the community if the authors could make the source code of this part available.

**Relation To Prior Work:**

Enough

**Summary And Contributions:**

This paper proposes a dataset called Multimodal C4 (mmc4). The proposed dataset is based on a c4 corpus containing only text. The authors added image data to c4 to create mmc4. Finally, the authors propose a mmc4 dataset containing 571M images, mmc4-core with stricter thresholds applied, and mmc4-ff/mmc4-core-ff with facial information removed.

---

> ### Author Response · Authors · 2023-08-17
> **Rebuttal Response to Reviewer GHE4**
>
> Thank you for your insightful comments and observations regarding our dataset! Here are our responses regarding your questions:
>
> > **Source code for curation**
>
> We already provide code to do bipartite matching, deduplication commands, etc., both for replicability and for future curation efforts. The scripts can be found at: https://github.com/allenai/mmc4. We are happy to provide additional source code, e.g., the face detection script for the camera ready.
>
> > **Supplementary submitted with main paper**
>
> We can rectify this in the camera ready as our main body is under 9 pages; thanks for pointing it out.

---

### Author Response · Authors · 2023-08-17
**Welcome Further Response and Discussion**

We thank all reviewers for their careful and constructive feedback. We were happy to see that mmc4 was well received!

We have posted the first round of rebuttal responses to each reviewer. Please let us know if we have addressed your concerns or if you have additional feedback or suggestions.

We highly appreciate your time and efforts and are looking forward to the discussions.

Best regards,

Authors of Submission#18

---

### Decision · Program_Chairs · 2023-09-22

**Decision:**

Accept (Poster)

**Comment:**

This paper introduces a MultiModal C4 (mmc4) dataset, with billion-scale corpus of images interleaved with text. This type of interleaved image-text data is lacking in the field and the scale of the dataset is significantly larger than previous multimodal datasets.The reviewers all agree that mmc4 could benefit future research on multimodal models. The authors also provide inital evidence by training OpenFlamingo on mmc4 that achieves promising results.